# Position: GenAI Systems Should Implement Contribution-Aware Revenue Sharing for Data Providers

**Gengrui Zhang** [1]

## Abstract

GenAI systems, particularly LLMs, rely heavily on vast amounts of publicly available digital content as training data. A significant portion of this content is protected by copyright. While large-scale data scraping may be lawful under certain jurisdictions, the use of copyrighted works to generate outputs that compete with or replicate original creations raises unresolved legal, economic, and ethical concerns. In this position paper, we argue that data providers should be fairly compensated based on their measurable contribution to inference-time outcomes, rather than through coarse, one-time licensing or blanket agreements. We examine alternative perspectives on data ownership, fair use, and model training, and discuss why existing approaches fail to align incentives between GenAI developers and content creators. We further outline concrete roadmaps and calls for action for developing decentralized systems that enable contribution-aware revenue sharing, including mechanisms for attribution, accounting, and payout at scale. We argue that fair revenue distribution for data providers will not only help resolve ongoing legal disputes surrounding GenAI systems, but also foster a new era of collaboration, rather than competition, between model developers and data creators. By incentivizing the production and sharing of high-quality datasets, such mechanisms can ultimately accelerate the development of more robust, trustworthy, and socially sustainable GenAI systems.

[1]Department of Electrical and Computer Engineering, Concordia University, Montreal, Canada. Correspondence to: Gengrui (Edward) Zhang <gengrui.zhang@concordia.ca>.

*Proceedings of the 43rd International Conference on Machine Learning*, Seoul, South Korea. PMLR 306, 2026. Copyright 2026 by the author(s).

## 1. Introduction

The rapid success of generative AI (GenAI) systems, particularly large language models (LLMs), has been driven by training on internet-scale data. These models treat publicly available digital content as raw material for building powerful generative capabilities. However, a substantial portion of this content is protected by copyright law. While large-scale data scraping may in some cases be lawful, the use of copyrighted works to generate outputs that compete with or closely replicate original creations raises significant legal, economic, and ethical challenges (Rane, 2025).

In the United States, this tension is shaped by the relatively flexible doctrine of fair use, which has introduced substantial legal uncertainty for GenAI developers. A growing number of high-profile class action lawsuits illustrate this uncertainty, including stock photography (Getty Images v Stability AI, 2025), news content (NYT v. Microsoft, 2023-12), and film characters (Disney v. Midjourney, 2025). Together, these cases underscore an increasingly adversarial relationship between AI developers and copyright holders, with courts often requiring plaintiffs to demonstrate concrete monetary harm while leaving broader questions unresolved.

In Canada, the governing principle is fair dealing, which permits the use of copyrighted works without authorization only for narrowly defined purposes such as research, education, criticism, review, parody, satire, and news reporting (Copyright Act, 1985). Compared to U.S. fair use, fair dealing provides less flexibility, raising further uncertainty about the legality of large-scale data use for GenAI training and deployment.

In Europe, recent regulatory developments further complicate the landscape. While text-and-data-mining exceptions exist, they are often coupled with opt-out mechanisms that allow rights holders to restrict use of their content. In parallel, the EU AI Act introduces new transparency obligations for providers of general-purpose AI models (EU Commission, 2025b). Together, these developments reinforce the principle that copyright holders retain control over how their works are used, creating a fragmented and increasingly restrictive regulatory environment for GenAI systems operating at global scale.

At the core of these disputes is the fact that **authors, artists, and other copyright holders are generally not compensated when their works are used to train GenAI systems**. The conflict carries wide-ranging implications, spanning social, cultural, legal, and policy dimensions: it undermines and discourages investment in high-quality content creation, threatens the livelihoods of creative professionals whose works risk being devalued or displaced by GenAI, exposes gaps in existing copyright frameworks governing rapidly evolving AI practices. Policymakers face a dilemma: on one hand, *they must foster AI innovation to remain globally competitive*; on the other, *they must protect creators, uphold copyright law, and maintain fairness in cultural and economic ecosystems* (Lucchi, 2024). Scholars have suggested remedies such as monetary funds to compensate creators (Samuelson, 2024; Hazra et al., 2025). However, the fundamental challenge remains unresolved: it is still unclear how such compensation mechanisms can be implemented in a way that is both transparent and fair, while also being auditable and adaptable to diverse real-world applications (Kandpal & Raffel, 2025).

**Our position. We argue that GenAI systems should be required to implement contribution-aware, inference-time revenue sharing mechanisms that compensate data providers proportionally to their measurable impact on model outputs, rather than relying on one-time licensing, blanket agreements, or uncompensated data use.** We contend that value generated by GenAI systems materializes at inference time and should be shared accordingly with those whose data meaningfully contributes to that value.

Any effective compensation framework should therefore satisfy three essential criteria: (1) compensation must be proportional to a data provider's contribution to inference-time outcomes (e.g., model outputs); (2) revenue distribution must be verifiable and auditable by all participating parties; and (3) the framework must be versatile enough to support a wide range of GenAI applications and deployment settings. In this context, we argue that a **decentralized system** is preferable to purely centralized approaches, as decentralization strengthens transparency, enables public or delegated auditing, and avoids lock-in to a single GenAI platform, thereby improving adaptability and long-term sustainability.

While prior work and public discourse have acknowledged the need to compensate data providers, most proposals stop short of articulating a clear position for plausible, technically grounded steps for achieving fair revenue sharing at scale. In contrast, this paper **advances an explicit position and outlines a concrete system framework** for implementing decentralized, contribution-aware compensation in GenAI ecosystems. We further identify actionable steps at the technical, community, and regulatory levels to enable adoption without slowing down the development of GenAI systems.

The remainder of this paper is organized as follows. Section 2 examines the current state of practice and the risks posed by existing compensation approaches. Section 3 articulates our core position statements in detail. Section 4 presents a system framework and architecture for realizing fair, contribution-aware revenue sharing. Section 5 discusses credible alternative views and explains their limitations. Finally, Section 6 outlines concrete calls to action for researchers, practitioners, and policymakers.

## 2. Reality and Risks

This section examines the current state of data compensation practices in GenAI systems and the systemic risks that arise from existing approaches. We first summarize prevailing realities observed in practice, drawing on recent legal, commercial, and regulatory developments. We then discuss the longer-term risks these realities pose to the sustainability, legality, and scalability of GenAI ecosystems.

### 2.1. The Reality

Despite sustained debate across legal, technical, and policy communities, the current landscape of data compensation in GenAI systems can be distilled into three persistent realities.

**Reality 1: In most cases, data providers are not paid at all.** For the majority of internet-scale training, the dominant practice remains uncompensated ingestion of publicly accessible content. The resulting conflict has escalated into major litigation, including the ongoing lawsuit brought by *The New York Times* against Microsoft and OpenAI, which alleges unpermitted copying and use of Times content for training and output generation (NYT v. Microsoft, 2023-12; NYT v. OpenAI, 2024). Similar tensions are visible in other domains and jurisdictions, such as *Getty Images v. Stability AI* in the UK, which directly tests how copyright and related rights apply to AI model development and outputs (Getty Images v Stability AI, 2025). These disputes illustrate that the status quo is not a stable equilibrium: creators increasingly contest uncompensated use, while developers face growing legal and reputational exposure.

**Reality 2: When compensation happens, it is typically coarse-grained and deal-driven.** A second pattern has emerged in which a small number of large rightsholders negotiate licensing or partnership agreements with GenAI model providers. This trend continued and expanded in 2025, with a growing number of bilateral agreements reported across the media and technology sectors (Digiday, 2026). For example, Axios entered a multi-year content and technology partnership with OpenAI aimed at supporting local journalism (OpenAI & Axios Partnership, 2025), while the Associated Press licensed real-time news content to Google for use in its Gemini models (Google Gem-

ini, 2025). Outside the U.S., Agence France-Presse (AFP) signed a multi-year licensing agreement with Mistral AI, granting access to its news corpus (Agence France-Presse, 2025). These agreements may provide meaningful revenue for a small set of large publishers, but they do not constitute a general solution: they are negotiated on a case-by-case basis, lack standardized or transparent terms, and provide little coverage for the long tail of creators and smaller data providers. As a result, licensing-based compensation remains fragmented and poorly aligned with the ongoing, inference-time value generated by GenAI systems.

**Reality 3: There is no widely adopted, transparent mechanism for scalable compensation.** Even where the normative goal of "paying creators" is accepted, there is no broadly deployed mechanism that can (i) attribute model behavior to specific training sources at inference time, (ii) account for usage at scale, and (iii) distribute revenue in a verifiable manner. Although emerging regulations have started to acknowledge this gap (e.g., the EU AI Act requires certain AI systems to disclose summaries of training data sources (EU Commission, 2025a)), such efforts do not themselves prioritize fair compensation. Consequently, current compensation mechanisms, when they exist, remain fragmented, opaque, and structurally biased toward large negotiating entities.

## 2.2. Risks and Challenges

The current realities create systemic risks for GenAI development and for the broader information ecosystem. These risks compound over time as GenAI systems scale in capability, deployment scope, and economic impact.

**Risk 1: GenAI may face a data access and sustainability crunch.** If uncompensated use continues, content platforms and publishers have incentives to restrict access, raise API prices, or explicitly prohibit use of their data for training. For example, Reddit announced major changes to its API access model, reflecting a broader shift toward monetization and access control (Lomas, 2023). More recently, X updated its developer terms to restrict third-party use of platform content for training AI models (Mehta, 2025; Staff, 2025). In parallel, technical opt-out mechanisms are evolving (e.g., robots-based approaches and opt-out protocols), which further reduces the pool of accessible training data. These trends suggest that the era of "free" internet-scale training data may be ending, and that sustainable access will increasingly depend on compensation and governance.

**Risk 2: The highest-value data is becoming the most contested.** It is widely recognized that high-quality data sources (e.g., professional journalism, curated archives, books, and specialized corpora) are particularly valuable for the performance of GenAI systems (e.g., the capability and reliability of LLMs) (Priestley et al., 2023; Hiniduma et al., 2025). High-quality data are also the most likely to be protected

by strong rights enforcement or paywalls (Villalobos et al., 2024; Duan et al., 2025). Data scarcity analyses project that, under continued scaling trends, availability of high-quality human-generated text could become a binding constraint for model development within the 2026–2032 horizon (Villalobos et al., 2022). As value concentrates in contested sources, compensation disputes become not just ethical or legal issues, but strategic bottlenecks affecting model quality and competitiveness (Liu et al., 2025).

**Risk 3: Legal uncertainty creates deployment and investment risk.** Legal and policy makers face a fundamental dilemma: on the one hand, *they must foster AI innovation to remain globally competitive*; on the other, *they must protect creators, uphold copyright law, and preserve fairness in cultural and economic ecosystems* (Lucchi, 2024). The lack of clear, settled doctrine exacerbates this tension. Ongoing litigation and unresolved legal questions, such as the boundaries of fair use and fair dealing, the status of output regurgitation and derivative works, and the legality of large-scale data acquisition practices, create substantial uncertainty for GenAI deployment, partnerships, and long-term investment.

High-profile cases illustrate how this uncertainty persists over extended time horizons. For example, key claims in *The New York Times* lawsuit against OpenAI and Microsoft have survived early procedural challenges, leaving material legal exposure unresolved for multiple years (NYT v. Microsoft, 2023-12). Similar frictions extend beyond text-based models, as image-generation disputes such as *Getty Images v. Stability AI* continue to test how copyright applies to model training and generative outputs (Getty Images v Stability AI, 2025). Together, these cases underscore how legal ambiguity translates directly into operational and financial risk for GenAI systems at scale.

**Risk 4: Global-scale expansion amplifies compliance complexity and fragmentation.** GenAI systems operate globally, but legal regimes vary substantially across jurisdictions. In Europe, for example, transparency obligations for general-purpose AI models and evolving copyright constraints imply compliance and reporting burdens that may not apply elsewhere, while opt-outs and access controls can differ across regions (Commission, 2025; Quintais, 2025). Fragmentation increases the cost of safe deployment and incentivizes opaque practices (e.g., undisclosed data mixtures), which in turn undermines trust (Luna et al., 2024; Kyrychenko et al., 2025). Without auditable, contribution-aware compensation mechanisms, global compliance becomes an arms race of legal risk management rather than a scalable, principled ecosystem (EU Commission, 2025b).

Taken together, these realities and risks motivate the need for contribution-aware revenue sharing frameworks that are scalable, verifiable, and interoperable across GenAI platforms and jurisdictions.

## 3. Our Positions

We take the position that fair and sustainable deployment of GenAI systems requires a fundamental shift in how data providers are compensated. Specifically, compensation must be tied to measurable contribution at inference time, proportional to the economic value generated by GenAI systems, and enforced through transparent, auditable mechanisms that are supported by both the ML community and legal authorities. Below, we articulate four concrete positions that together define this framework.

> **Position 1:** *Data providers should be compensated proportionally to their impact on inference-time outcomes.*

We argue that compensation should be based on the measurable impact of a data provider's contributions on inference-time outputs, rather than on coarse proxies such as dataset size, training-time inclusion, or one-time licensing fees (Samuelson, 2025; Mattila, 2025). The value of data in GenAI systems is ultimately realized during inference, when models generate outputs that users consume and monetize. Compensation mechanisms that ignore inference-time impact fail to reflect where value is actually created.

Recent advances in data attribution and influence estimation provide a foundation for quantifying such impact (Hammoudeh & Lowd, 2024). Techniques such as influence functions, gradient-based attribution, data valuation methods, and emerging systems like training data tracking and influence tracing (e.g., TracIn (Pruthi et al., 2020), UnTrack (Isonuma & Titov, 2024), and DataInf (Kwon et al., 2023)) enable estimation of how individual samples or datasets affect model predictions and outputs (Sim et al., 2022b). While these methods are imperfect and often approximate, they represent a critical step toward operationalizing contribution-aware compensation.

Importantly, proportional compensation does not require exact attribution for every token or output (Ko et al., 2024). Approximate, probabilistic, or amortized attribution, when transparently defined and consistently applied, can still provide a fairer alternative to today's binary "paid vs. unpaid" regime (Ghorbani & Zou, 2019; Tarun et al., 2024). We contend that the absence of perfect attribution should not be used as a justification for ignoring contribution altogether. Instead, improving attribution accuracy should be treated as an ongoing research and engineering goal, analogous to continual improvements in model evaluation or robustness.

> **Position 2:** *Compensation should scale with GenAI-generated revenue, including subscriptions and advertising income.*

We further argue that compensation to data providers should be proportional to the revenue generated by GenAI systems, rather than fixed or capped payments disconnected from downstream economic success. GenAI models increasingly operate as revenue-generating products, earning income through subscriptions, usage-based pricing, enterprise licensing, and advertising. Data providers contribute to the quality, reliability, and attractiveness of these systems, and should therefore share in their financial upside.

For subscription-based revenue streams, contribution can be estimated using user satisfaction signals, retention metrics, or platform-specific evaluation algorithms. Rather than attributing revenue to isolated outputs, contributions can be aggregated over longer interaction sessions, capturing user-perceived value such as satisfaction and retention. In multi-turn interactions, attribution can be aggregated at session level instead of assigning equal weight to intermediate responses, aligning compensation with final utility.

This principle is especially important in **advertising-based revenue streams**, where user engagement and content quality directly translate into revenue. Currently, ads-based revenue dominates the income of major internet companies (e.g., approximately 77% of Google's and 98% of Meta's revenue originates from ads). In such settings, high-quality GenAI outputs derived from valuable training data can increase user retention, click-through rates, conversions, and overall advertising value. Flat licensing agreements or one-time payments fail to capture this continuing and compounding contribution over time.

Revenue-proportional compensation creates stronger incentive alignment across stakeholders. Model developers are encouraged to improve attribution accuracy and transparency, data providers are motivated to produce and share higher-quality content, and users ultimately benefit from improved GenAI outputs (Tang et al., 2025). In contrast, compensation schemes decoupled from downstream value risk reinforcing existing tensions, where creators bear long-term costs while developers capture compounding economic benefits (Azcoitia & Laoutaris, 2022).

This position may be challenged by arguments that revenue sharing introduces excessive complexity or discourages innovation. We argue that, although contribution-aware revenue sharing requires estimating training data influence, the associated system complexity can be managed through standardized accounting, approximate attribution, and automated payout mechanisms, and therefore should not hinder the development of GenAI systems. Moreover, influence estimation can be performed asynchronously or offline, decoupled from the production inference pipeline. As more efficient attribution methods continue to emerge (Pruthi et al., 2020; Kwon et al., 2023), pursuing this position encourages a more sustainable ecosystem that rewards long-term value creation rather than short-term exploitation of freely available data (Parker et al., 2017).

**Position 3:** *Compensation processes and outcomes should be auditable and verifiable by all stakeholders.*

A compensation framework that cannot be independently audited is unlikely to earn trust or achieve broad adoption. We therefore take the position that both the process of attribution and the resulting compensation outcomes should be auditable and verifiable by all relevant parties, including data providers, GenAI model operators, and selected or delegated third-party auditors.

Auditability serves several purposes. It deters malicious behavior such as underreporting revenue, manipulating attribution metrics, or selectively excluding contributors. It enables dispute resolution by providing evidence that compensation rules were applied correctly. It also supports regulatory compliance across jurisdictions with differing transparency and accountability requirements.

We argue that decentralized verification mechanisms are particularly well suited to this task. While centralized systems may be simpler to deploy initially, they concentrate trust in a single entity and create incentives for opaque or biased accounting. In contrast, auditable logs, verifiable computation, and programmable enforcement mechanisms (e.g., smart contracts) enable public or semi-public verification without requiring universal trust in a single operator.

Crucially, auditability does not imply full disclosure of proprietary model details or training data (Pruthi et al., 2020; Kwon et al., 2023). Well-designed systems can balance transparency with confidentiality by exposing only what is necessary to verify correctness and fairness. The goal is not radical openness, but credible verifiability.

**Position 4:** *Community and legal support are necessary for real-world incremental adoption*

Finally, we argue that contribution-aware compensation frameworks cannot succeed through technical innovation alone. Broad adoption requires support from the ML research community, industry stakeholders, and legal and regulatory authorities (Raji et al., 2020).

Within the ML community, norms must evolve to recognize compensation as a core systems and governance problem rather than a peripheral ethical concern. Benchmarking attribution methods, sharing best practices, and incorporating compensation considerations into system design discussions are necessary steps toward normalization.

From a legal and policy perspective, regulators should provide clarity and incentives for adopting transparent compensation mechanisms. Legal recognition of contribution-aware compensation can reduce uncertainty for developers and creators alike, lowering barriers to experimentation and deployment. Importantly, supportive regulation can enable

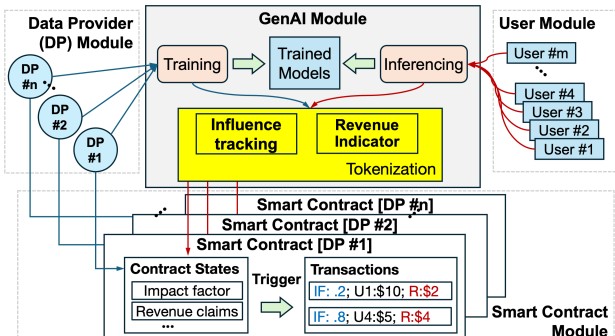

*Figure 1.* Overview of a decentralized framework for fair GenAI revenue compensation, integrating influence tracking, tokenized revenue indicators, and smart contracts for auditable, revenue-proportional payouts.

innovation by replacing adversarial legal battles with structured, auditable solutions (Veale & Binns, 2024).

We emphasize that this initiative is not intended to slow down the development of GenAI systems. On the contrary, by reducing legal risk, aligning incentives, and stabilizing access to high-quality data, fair compensation frameworks can accelerate sustainable innovation. Moreover, by rewarding contributors in proportion to the value their data generates, **such frameworks incentivize data providers to produce, curate, and share higher-quality training data**, directly improving model performance, reliability, and the pace of future GenAI advancement.

## 4. Solutions for Contribution-Aware Revenue Sharing in GenAI Systems

We envision a two-layer system framework for revenue compensation in GenAI ecosystems. The goal of this framework is to enable **contribution-aware, revenue-proportional, and auditable compensation** for data providers. At a high level, the system consists of two tightly coupled layers (shown in Figure 1): (1) influence tracking and tokenization embedded within GenAI systems, and (2) decentralized smart contracts that enforce fair and auditable revenue sharing. Together, these components form a unified data governance layer that bridges model behavior, economic value, and verifiable compensation.

In the proposed framework, data providers, GenAI service operators, and users jointly participate in a governance model. Data providers contribute training data to GenAI systems; users generate inference requests that produce economic value through subscriptions, usage fees, or advertising; and GenAI operators coordinate model training and inference. Rather than treating compensation as an external legal or contractual process, the **framework embeds compensation logic directly into the GenAI lifecycle**.

## 4.1. COMPONENT 1: Influence tracking and tokenization in GenAI systems

The objective of the first component is to quantify how training data contributed by different data providers affects inference-time outcomes, and to translate this influence into standardized impact factors.

We propose to leverage influence tracking techniques, such as gradient-based influence estimation (Pruthi et al., 2020) and training-data tracing mechanisms (Kwon et al., 2023), and extend them to GenAI-scale systems. These techniques estimate the relative impact of individual data samples during inference with quantified influence scores. Such **attribution can be computed periodically and offline**, decoupled from the production inference pipeline to avoid interference with normal serving workloads.

**Step 1: Tokenizing training data impact.** Influence scores are converted into digital influence tokens that represent a data provider's relative contribution to inference outcomes. These tokens abstract away model-specific details and provide a standardized interface between GenAI systems and decentralized accounting infrastructure.

Tokenization decouples attribution logic from compensation logic, allowing attribution methods to evolve without redesigning the payment system. It also enables interoperability across models, platforms, and deployment settings by expressing contribution in a common, verifiable format.

**Step 2: Tokenizing impact with respect to revenue indicators.** The proposed framework associates influence tokens with revenue indicators, such as subscription payments, per-query fees, or advertising income. As such, when a user interaction generates revenue, a portion of that revenue is allocated proportionally based on the influence tokens associated with the inference outputs involved.

This design ensures that compensation scales with real economic outcomes rather than static assumptions. Data providers whose contributions consistently improve high-value outputs receive proportionally higher compensation, aligning incentives across stakeholders.

## 4.2. COMPONENT 2: Smart contracts for fair and auditable revenue sharing

The second component implements compensation enforcement using smart contracts on blockchains. These contracts act as neutral, programmable arbiters that manage revenue claims, execute payouts, and expose auditable records.

**Step 1: Revenue sharing contracts for GenAI-specific revenue sharing.** Revenue sharing should be governed through digital contracts between data providers and GenAI systems. These contracts are predefined and agreed upon before inference begins, and are automatically triggered by revenue-generating inference events. Each contract is instantiated per data provider and maintains state information such as accumulated influence tokens, revenue claims, and payout histories.

When revenue events occur, the contracts are triggered using influence tokens and revenue indicators generated by the GenAI system. Payouts are then computed automatically according to predefined and publicly verifiable rules. This removes discretionary control from any single party and ensures that compensation follows agreed-upon logic transparently.

The system architecture is flexible as long as transparency and auditability are preserved. We envision three evolving deployment models:

- **Centralized systems:** GenAI providers centrally manage digital contracts with distributed verification mechanisms with data providers.

- **Semi-decentralized systems:** GenAI providers and data providers jointly maintain distributed databases with auditing capabilities.

- **Fully decentralized systems:** GenAI providers, data providers, and auditing parties coordinate through peer-to-peer and self-sustained contract management (e.g., smart contracts).

For fully decentralized deployments, we advocate permissioned blockchains as one practical solution. Such systems enable jointly managed agreements among data providers, GenAI operators, auditors, and regulatory observers while tolerating Byzantine faults (Zhang et al., 2024). In addition, permissioned blockchains can obtain higher throughput and lower latency (Androulaki et al., 2018), making them suitable for high-throughput GenAI environments.

**Step 2: Auditability and community oversight.** All contract states are auditable by authorized participants. Data providers can verify that their contributions were accounted for correctly, GenAI operators can demonstrate compliance, and community members and legal authorities can monitor the system in real time (Liu et al., 2019).

By integrating influence tracking, tokenization, and decentralized smart contracts, the proposed framework enables fair, revenue-proportional, and auditable compensation for GenAI data providers. It is worth noting that this design does not slow down GenAI inference or deployment: blockchain operations occur after inference is completed and do not block or interfere with normal system usage.

### 4.3. Examples

An example of the revenue-sharing workflow for ads-based income stream proceeds as follows (shown in Figure 1).

① Users U1 and U4 both ask the LLM (GenAI module): *"What should I wear to a cocktail party?"* The model generates recommendations for different events and provides links to purchase suggested clothing items.

② Influence tracking shows that Data Provider 1 (DP1) contributed 20% to the output for U1 and 80% to the output for U4. These impact factors are recorded as states in the smart contracts.

③ Both U1 and U4 follow the links and purchase clothing, generating advertisement revenue for the LLM, with $10 and $5 allocated for sharing with data providers, respectively. The revenue indicator updates the contract state and triggers new transactions.

④ Based on the impact factors, DP1 is entitled to $2 ($0.2 \times \$10$ from U1) and $4 ($0.8 \times \$5$ from U4).

⑤ DP1 executes and broadcasts the corresponding smart contract in the blockchain network, which is verified and validated by all the participating peers.

⑥ Once compensation is confirmed in the blockchain, the transaction is finalized, and DP1 receives the appropriate revenue share.

This architecture can also be adapted to subscription-based revenue distribution. In such settings, the revenue indicator may use user satisfaction signals, retention metrics, or platform-specific evaluation algorithms to estimate contribution. Instead of associating revenue with a single inference event, contributions can be aggregated across sessions or longer interaction horizons, producing contribution scores that are later used for revenue allocation.

More broadly, the proposed architecture provides a general-purpose framework and workflow for compensating data providers. Its components (e.g., influence estimation methods) are modular and replaceable, **making our framework method-agnostic** and capable of incorporating more efficient alternatives as they emerge. The proposed system incentivizes the creation and sharing of higher-quality training data, encourages collaboration between data providers and GenAI developers, and supports the development of more reliable and sustainable GenAI systems.

## 5. Alternative Views

Several alternative positions challenge the need for contribution-aware, inference-time compensation in GenAI systems. We discuss two prominent and credible views below and explain why we believe they are ultimately insufficient to support sustainable and fair GenAI ecosystems.

**Alternative View 1: Fair use permits uncompensated use of training data.** One widely held view is that artists and content providers should not be compensated at all,

because training GenAI models on publicly available data falls under the doctrine of fair use, particularly in the United States (Lemley & Casey, 2021; Henderson et al., 2023). Proponents of this view argue that model training is a transformative use of data, that models do not store or reproduce training examples verbatim in most cases, and that requiring compensation would impose prohibitive costs that slow innovation. From this perspective, existing copyright frameworks already strike an appropriate balance between innovation and creator rights, and no new compensation mechanisms are necessary (Lee, 2025).

This position is not without merit. In several early-stage rulings, courts have allowed GenAI developers to proceed by emphasizing the transformative nature of model training and the fact-bound nature of fair use analysis. For instance, in (Bartz v. Anthropic PBC, 2025), a U.S. District Court held in June 2025 that the use of copyrighted works to train AI models qualified as fair use, focusing on the transformative purpose of training even while distinguishing non-fair use aspects of data acquisition. Two days later, in a separate case involving Meta's AI models, a federal judge similarly found that the record before the court supported a fair use finding but explained that plaintiffs' failure to present strong evidence on market harm was central to the outcome (Kadrey v. Meta Platforms, 2025). These early decisions apply traditional fair use factors on a highly case-specific basis and show that demonstrating concrete market harm remains a key challenge for copyright owners (Ropes & Gray, 2025; Crowell & Moring, 2025).

However, the absence of demonstrated harm at this early stage does not imply that economic harm will not emerge in the future. As GenAI systems expand and increasingly substitute for human creators in domains such as journalism, illustration, and creative writing, revenue displacement may become clearer and more measurable over time (Hullman et al., 2023; Porquet et al., 2025). Under such conditions, future legal assessments may differ from current preliminary rulings (American Action Forum, 2025).

Moreover, reliance on fair use alone is not sustainable across jurisdictions. Legal frameworks in Canada (ISED Canada, 2024) and Europe provide more limited exceptions for data use, and increasingly emphasize rights-holder control and transparency (EU Commission, 2024). Exclusive dependence on fair use therefore fails to provide a stable, globally applicable foundation for GenAI development. From a systems perspective, uncompensated substitution also risks undermining the production of high-quality data, ultimately constraining future model improvement and creating a lose–lose outcome for both creators and GenAI developers.

Our position does not reject fair use outright. Rather, we argue that fair use alone is an unstable foundation for long-term GenAI ecosystems (Kyi et al., 2025). Contribution-

aware compensation offers a complementary mechanism that internalizes economic externalities, aligns incentives between developers and data providers, and reduces legal resources to resolve complex valuation questions.

**Alternative View 2: One-time licensing is sufficient compensation.** A second alternative view holds that licensing-based compensation, typically through fixed payment agreements, is sufficient (U.S. Copyright Office, 2025; Axhamn, 2024). Under this model, GenAI developers negotiate licenses with data providers or intermediaries, pay an upfront fee, and are thereafter free to use the data without ongoing obligations (Chesterman, 2025). This approach is appealing because it is simple, contractually familiar, and avoids the complexity of attribution, auditing, and revenue sharing (Sim et al., 2022a).

Licensing-based models have indeed emerged in practice, particularly involving large publishers and platform-scale content owners (e.g., licensing framework (Prasad & Padilla, 2025) and collective licensing (Samuelson, 2025; Mattila, 2025)). These agreements demonstrate that compensation is possible and that markets can form around GenAI data access. However, we argue that one-time licensing fails to address several structural problems.

First, fixed payments are poorly aligned with the ongoing and potentially compounding value generated by GenAI systems. Data that contributes disproportionately to high-revenue outputs is treated the same as marginal data, leading to systematic mispricing. Second, licensing regimes favor large, centralized actors who can negotiate at scale, while excluding smaller creators and long-tail data providers. Third, opaque licensing terms make it difficult for external parties to assess fairness, audit compliance, or adapt agreements as models and revenue streams evolve (Abnar et al., 2021).

In contrast, contribution-aware, revenue-proportional compensation directly ties payment to realized value and adapts automatically as GenAI systems scale. While more complex, this approach better reflects how value is created and distributed in modern AI systems (Wang et al., 2024).

Both alternative views offer partial solutions, but neither adequately addresses incentive alignment, long-term sustainability, and transparency at GenAI scale. Our position argues that these limitations motivate the need for contribution-aware, auditable compensation mechanisms that evolve alongside GenAI systems rather than relying on static legal or contractual assumptions.

# 6. Call to Action

Realizing fair, contribution-aware revenue sharing in GenAI systems requires coordinated action across technical research, the machine learning community, and regulatory

and institutional stakeholders. The decentralized framework outlined in Section 4 demonstrates that such systems are technically plausible; the remaining challenge is to translate this vision into practice. We therefore outline concrete steps at each level to move from position to deployment.

## 6.1. At the technical level

The ML and systems research communities should advance the two crucial components introduced in Section 4. The following calls highlight key technologies and research directions needed to make such systems practical and scalable.

> **Call 1:** *Develop more efficient and scalable influence estimation methods.*

In particular, efficient techniques for identifying the top-$k$ most influential training samples for a given model output are essential. Existing approaches to influence estimation, attribution, and data valuation are often computationally expensive at GenAI scale, limiting their practical adoption (Pruthi et al., 2020; Kwon et al., 2023). Progress in approximation techniques, amortized attribution, and system-level optimizations is therefore necessary to make contribution-aware compensation viable in real-world deployments (Isonuma & Titov, 2024; Sim et al., 2022b).

Future influence estimation methods should support multiple levels of attribution granularity, such as token-level, sample-level, and group-level influence. Different granularities introduce different trade-offs among interpretability, attribution precision, and computational efficiency, and future systems should evaluate their practical feasibility under large-scale GenAI workloads.

It is also important to recognize the inherent **trade-off between attribution accuracy and system performance**. Many influence estimation methods rely on gradient-based comparisons between training samples and inference queries. Techniques that reduce computational overhead (e.g., dimensionality reduction or random projection methods) can significantly improve scalability, but may also reduce attribution fidelity. Designing efficient methods that balance scalability, precision, and robustness remains an important open research challenge.

> **Call 2:** *Building influence estimation under synthetic data and multi-stage GenAI training pipelines.*

We note that synthetic data generation is increasingly performed within GenAI workflows by model operators (e.g., during distillation (Jazbec et al., 2024; Bucilua et al., 2006)), in which case it is already part of the system and does not introduce additional compensation requirements. However, synthetic data does not eliminate contribution; rather, it propagates contribution across stages.

For externally generated synthetic data, contributions can still be tracked through data lineage and provenance mechanisms. In addition, governance at the data ingestion and provenance layer (e.g., requiring source disclosure or restricting untracked synthetic data) can make circumvention more detectable and auditable.

Future influence estimation methods should also support complex multi-stage training pipelines, including pretraining, fine-tuning, distillation, and RLHF, where contribution propagates across intermediate models and datasets.

> **Call 3:** *Explore efficient, portable, auditable, and fault-tolerant decentralized contracting infrastructures for GenAI systems.*

More efficient decentralized infrastructure is needed to support transparent accounting and revenue distribution. This includes advances in scalable blockchain and distributed ledger technologies capable of handling high-throughput attribution records, low-latency settlement, and verifiable auditing. Equally important are incentive mechanisms that encourage sustained participation by data providers, model developers, and independent auditors without introducing excessive overhead or security risks.

Tighter integration between GenAI platforms and decentralized infrastructures should also be explored. GenAI systems are typically governed in a centralized manner, while revenue-sharing contracts may operate in decentralized environments. These two components should therefore interoperate seamlessly. For example, decentralized smart contracts (e.g., blockchain-based solutions) can automate attribution-based payouts, enforce revenue-sharing policies, and reduce reliance on trusted intermediaries. Designing secure, interoperable, and fault-tolerant interfaces between GenAI inference pipelines and decentralized execution environments remains an important open systems challenge.

### 6.2. At the AI community level

Beyond individual technical advances, the ML community should work toward building an ecosystem that normalizes contribution-aware data usage.

> **Call 4:** *Foster ecosystem support and community standards for contribution-aware GenAI systems.*

This includes developing shared benchmarks for data attribution, publishing datasets with richer provenance and metadata, and establishing standard interfaces for reporting and verifying data contributions. GenAI community norms should evolve to treat fair compensation not as an optional ethical consideration, but as a core design principle.

Equally important is the collective willingness of GenAI developers and researchers to acknowledge the economic value of training data and data providers. Without such commitment, technical solutions alone are unlikely to achieve meaningful adoption or long-term impact.

### 6.3. At the regulatory and institutional level

Regulators and policymakers should encourage, and where appropriate require, GenAI developers to adopt transparent and verifiable compensation mechanisms for data providers. Rather than prescribing rigid technical implementations, regulatory frameworks should focus on outcome-oriented principles such as auditability, proportional compensation, transparency, and protection of data providers' rights.

> **Call 5:** *Develop adaptive regulatory and institutional frameworks that evolve alongside technical progress and actively support fair compensation for data providers in GenAI systems.*

Policy and regulation should adapt alongside rapid advances in GenAI technologies and attribution methods, rather than remaining static while technical capabilities evolve. Early-stage policies may initially rely on lightweight or approximate attribution mechanisms, while future frameworks can incorporate more precise and contribution-aware systems as the technology matures. Importantly, regulatory efforts should remain aligned with the broader goal of ensuring fair and sustainable compensation for data providers rather than reinforcing existing imbalances in bargaining power.

In addition, clear legal frameworks are needed to define the rights and obligations of GenAI developers and data providers in contribution-aware revenue-sharing systems. Public institutions should also play a role in supervising and auditing deployed systems to ensure compliance, deter malicious behavior, and maintain a fair competitive landscape. Together, these measures can help align innovation incentives with legal and societal expectations, enabling more sustainable and collaborative GenAI ecosystems.

## 7. Conclusion

GenAI systems generate substantial economic value from large-scale training data, yet current compensation practices remain coarse, opaque, and misaligned with how value is produced at inference time. We argued that sustainable GenAI ecosystems require contribution-aware, revenue-proportional, and auditable compensation for data providers. By integrating influence tracking, tokenization, and decentralized enforcement, such frameworks can align incentives, reduce legal uncertainty, stabilize access to high-quality data, and accelerate responsible GenAI development rather than hinder it. The central challenge going forward is not whether compensation is necessary, but how to implement it in a scalable, verifiable, and interoperable manner that supports continued innovation.

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

Copyright Act. Department of Justice Canada, Copyright Act, RSC 1985, c C-42, s 27.1, 1985.

Crowell & Moring. Ai companies prevail in path-breaking decisions on fair use. Crowell Client Alerts, 2025.

Digiday. A timeline of the major deals between publishers and ai companies in 2025. *Digiday*, 1 2026. URL https://digiday.com/media/a-timeline-of-the-major-deals-between-publishers-and-ai-tech-companies-in-2025/.

Disney v. Midjourney. U.S. District Court, Central District of California. *No. 2:25-cv-05275*, 2025.

Duan, F., Zhang, X., Wang, S., Que, H., Liu, Y., Rong, W., and Cai, X. Enhancing llms via high-knowledge data selection. In *Proceedings of the AAAI Conference on Artificial Intelligence*, volume 39, pp. 23832–23840, 2025.

EU Commission. The eu artificial intelligence act, 2024.

EU Commission. Commission presents template for general-purpose ai model providers to summarise the data used to train their model, 2025a.

EU Commission. Contents of the code of practice on generative ai. https://digital-strategy.ec.europa.eu/en/policies/contents-code-gpai, 2025b. European Union policy outlining the governance and transparency expectations for general-purpose AI models.

Getty Images v Stability AI. Getty images v stability ai: The uk courts' first word on use of copyright works in ai model development. 11 2025.

Ghorbani, A. and Zou, J. Data shapley: Equitable valuation of data for machine learning. In *International conference on machine learning*, pp. 2242–2251. PMLR, 2019.

Google Gemini. Working with The Associated Press to provide fresh results for the Gemini APP. https://blog.google/products-and-platforms/products/news/associated-press-gemini-app/, Jan 2025.

Hammoudeh, Z. and Lowd, D. Training data influence analysis and estimation: A survey. *Machine Learning*, 113(5):2351–2403, 2024.

Hazra, S., Majumder, B. P., and Chakrabarty, T. Position: AI safety should prioritize the future of work. In *Forty-second International Conference on Machine Learning Position Paper Track*, 2025. URL https://openreview.net/forum?id=CA9NxmmUG5.

Henderson, P., Li, X., Jurafsky, D., Hashimoto, T., Lemley, M. A., and Liang, P. Foundation models and fair use. *Journal of Machine Learning Research*, 24(400):1–79, 2023.

Hiniduma, K., Byna, S., and Bez, J. L. Data readiness for ai: A 360-degree survey. *ACM Computing Surveys*, 57 (9):1–39, 2025.

Hullman, J., Holtzman, A., and Gelman, A. Artificial intelligence and aesthetic judgment. *arXiv preprint arXiv:2309.12338*, 2023.

ISED Canada. Copyright in the age of generative artificial intelligence: What we heard report, 2024.

Isonuma, M. and Titov, I. Unlearning traces the influential training data of language models. *arXiv preprint arXiv:2401.15241*, 2024.

Jazbec, M., Xia, M., Mallick, A., Madrigal, D., Han, D., Kessler, S., and Ruhle, V. On efficient distillation from llms to slms. In *NeurIPS 2024 Workshop on Fine-Tuning in Modern Machine Learning: Principles and Scalability*, 2024.

Kadrey v. Meta Platforms. Northern district of california judge rules that meta's use of copyrighted works to train ai models was fair use on the record before the court. Goodwin Law Insights, Jun 2025.

Kandpal, N. and Raffel, C. Position: The most expensive part of an llm should be its training data. *arXiv preprint*, 2025. URL https://arxiv.org/abs/2504.12427.

Ko, M., Kang, F., Jin, M., Yu, Z., and Jia, R. The mirrored influence hypothesis: Efficient data influence estimation by harnessing forward passes. In *Proceedings of the IEEE/CVF Conference on Computer Vision and Pattern Recognition*, pp. 26286–26295, 2024.

Kwon, Y., Wu, E., Wu, K., and Zou, J. Datainf: Efficiently estimating data influence in lora-tuned llms and diffusion models. *arXiv preprint arXiv:2310.00902*, 2023.

Kyi, L., Mahuli, A., Silberman, M. S., Binns, R., Zhao, J., and Biega, A. J. Governance of generative ai in creative work: Consent, credit, compensation, and beyond. In *Proceedings of the 2025 CHI Conference on Human Factors in Computing Systems*, pp. 1–16, 2025.

Kyrychenko, M., Mudryi, M., and Chaklosh, M. Global ai governance overview: Understanding regulatory requirements across global jurisdictions. *arXiv preprint arXiv:2512.02046*, 2025.

Lee, E. Fair use and the origin of ai training. *Houston Law Review*, 63(1):104, 2025.

Lemley, M. A. and Casey, B. Fair learning. *Texas Law Review*, 99(4):743–820, 2021. URL https://texaslawreview.org/fair-learning/.

Liu, M., Wu, K., and Xu, J. J. How will blockchain technology impact auditing and accounting: Permissionless versus permissioned blockchain. *Current Issues in auditing*, 13(2):A19–A29, 2019.

Liu, Y., Cao, J., Liu, C., Ding, K., and Jin, L. Datasets for large language models: A comprehensive survey. *Artificial Intelligence Review*, 58(12):403, 2025.

Lomas, N. Reddit will begin charging for access to its api. *TechCrunch*, 4 2023. URL https://techcrunch.com/2023/04/18/reddit-will-begin-charging-for-access-to-its-api/.

Lucchi, N. Chatgpt: A case study on copyright challenges for generative artificial intelligence systems. *European Journal of Risk Regulation*, 15(3):602–624, 2024.

Luna, J., Tan, I., Xie, X., and Jiang, L. Navigating governance paradigms: A cross-regional comparative study of generative ai governance processes & principles. In *Proceedings of the AAAI/ACM Conference on AI, Ethics, and Society*, volume 7, pp. 917–931, 2024.

Mattila, T. Extended collective licensing as a solution to copyright frictions in aidriven creative economy. *Available at SSRN 5815722*, 2025.

Mehta, I. X changes its terms to bar training of ai models using its content. *TechCrunch*, 6 2025. URL https://techcrunch.com/2025/06/05/x-changes-its-terms-to-bar-training-of-ai-models-using-its-content/.

NYT v. Microsoft. U.s. district court for the southern district of new york. *No. 1:23-cv-11195 (S.D.N.Y.)*, 2023-12.

NYT v. OpenAI. Nyt v. openai: The times's about-face. *Harvard Law Review Blog*, 4 2024.

OpenAI & Axios Partnership. Partnering with Axios expands OpenAI's work with the news industry, 1 2025. URL https://openai.com/index/partnering-with-axios-expands-openai-work-with-the-news-industry/.

Parker, G. G., Alstyne, M. W. V., and Choudary, S. P. Platform ecosystems: How developers invert the firm. *MIS Quarterly*, 41(1):255–266, 2017. Shows how revenue sharing and standardized payouts are foundational to digital platform ecosystems such as app stores and content platforms.

Porquet, J., Wang, S., and Chilton, L. B. Copying style, extracting value: Illustrators' perception of ai style transfer and its impact on creative labor. In *Proceedings of the 2025 CHI Conference on Human Factors in Computing Systems*, pp. 1–16, 2025.

Prasad, K. and Padilla, J. Generative ai models at the gate: Licensing frameworks for the effective and efficient protection of copyright protected content in an ai world. SSRN, May 2025.

Priestley, M., O'donnell, F., and Simperl, E. A survey of data quality requirements that matter in ml development pipelines. *ACM Journal of Data and Information Quality*, 15(2):1–39, 2023.

Pruthi, G., Liu, F., and Sundararajan, M. Estimating training data influence by tracing gradient descent. *Advances in Neural Information Processing Systems*, 2020.

Quintais, J. P. Generative ai, copyright and the ai act. *Computer Law & Security Review*, 56:106107, 2025.

Raji, I. D., Smart, A., White, R. N., Mitchell, M., Gebru, T., Hutchinson, B., Smith-Loud, J., Theron, D., and Barnes, P. Closing the ai accountability gap: Defining an end-to-end framework for internal algorithmic auditing. In *Proceedings of the 2020 conference on fairness, accountability, and transparency*, pp. 33–44, 2020.

Rane, S. Position: AI's growing due process problem. In *Forty-second International Conference on Machine Learning Position Paper Track*, 2025. URL https://openreview.net/forum?id=TEkyydR6il.

Ropes & Gray. A tale of three cases: How fair use is playing out in ai copyright lawsuits. Ropes & Gray Insights, Jul 2025.

Samuelson, P. How to think about remedies in the generative AI copyright cases. *Communications of the ACM*, 67(7): 27–30, 2024.

Samuelson, P. Assessing the feasibility of collective licensing of in-copyright works for generative ai training. SSRN, 2025.

Sim, R. H. L., Xu, X., and Low, B. K. H. Data valuation in machine learning: "ingredients", strategies, and open challenges. In *Proceedings of the Thirty-First International Joint Conference on Artificial Intelligence (IJCAI)*, 2022a.

Sim, R. H. L., Xu, X., and Low, B. K. H. Data valuation in machine learning:" ingredients", strategies, and open challenges. In *IJCAI*, pp. 5607–5614, 2022b.

Staff, T. V. X's new policy prevents companies from using posts to fine-tune or train ai models. *The Verge*, 6 2025. URL https://www.theverge.com/news/680626/x-ai-training-ban-posts.

Tang, Y., Liu, Y., and Liu, D. Data asset valuation model based on generative artificial intelligence. *PLoS One*, 20 (8):e0328926, 2025.

Tarun, A., Chundawat, V., Mandal, M., Tan, H. M., Chen, B., and Kankanhalli, M. Ecoval: An efficient data valuation framework for machine learning. In *Proceedings of the 30th ACM SIGKDD Conference on Knowledge Discovery and Data Mining*, pp. 2866–2875, 2024.

U.S. Copyright Office. Copyright and Artificial Intelligence, Part 3: Generative AI Training (Pre-Publication Version), May 2025.

Veale, M. and Binns, R. Regulating ai: Lessons from the eu ai act. *Computer Law & Security Review*, 2024.

Villalobos, P., Ho, A., Sevilla, J., Besiroglu, T., Heim, L., and Hobbhahn, M. Will we run out of data? limits of llm scaling based on human-generated data. *arXiv preprint arXiv:2211.04325*, 2022. URL https://arxiv.org/abs/2211.04325.

Villalobos, P., Ho, A., Sevilla, J., Besiroglu, T., Heim, L., and Hobbhahn, M. Position: Will we run out of data? limits of llm scaling based on human-generated data. In *Forty-first International Conference on Machine Learning*, 2024.

Wang, J. T., Deng, Z., Chiba-Okabe, H., Barak, B., and Su, W. J. An economic solution to copyright challenges of generative ai. *arXiv preprint*, 2024. URL https://arxiv.org/abs/2404.13964.

Zhang, G., Pan, F., Mao, Y., Tijanic, S., Dang'Ana, M., Motepalli, S., Zhang, S., and Jacobsen, H.-A. Reaching consensus in the byzantine empire: A comprehensive review of bft consensus algorithms. *ACM Computing Surveys*, 56(5):1–41, 2024.

