# OpenReview forum: "Position: GenAI Systems Should Implement Contribution-Aware Revenue Sharing for Data Providers"
_ICML.cc/2026/Position_Paper_Track — ICML 2026 Position Paper Track regular_

### Official Review · Reviewer_DaD3 · 2026-03-07

**Significance:** 4
**Argument Clarity:** 3
**Rating:** 4
**Confidence:** 4

**Questions:**

1. How would such systems measure influence in a computationally feasible way?
2. What is the granularity of the influence metric? (token-wise or sample-wise)?
3. How would biases be prevented?
4. The paper argues decentralization is more beneficial, but would the influence be measured in a centralized way?

**Alternative Views Section:**

Yes

**Compliance With Llm Reviewing Policy A Conservative:**

Affirmed.

**Discussion Potential:**

3

**Paper Summary:**

The paper provides the position of a contribution-aware compensation framework for data providers. The main motivation behind this argument is that copyrighted and valuable data from providers is used to train GenAI systems that may generate revenue due to these sources. The paper proposes utilizing data attribution techniques (specifically Influence estimation methods) to measure the provider's impact (using the influence score of a data source on the model's prediction) and then using a blockchain to distribute the revenue proportionately.

**Position:**

Yes

**Position In Title:**

Yes

**Related Work:**

3

**Strengths And Weaknesses:**

# Strengths
- The paper highlights all aspects of the position very well.
- The topic of using a fair compensation system is very relevant.

# Weaknesses
- The concept of the contribution of the data provider is not specified well. For example, at what granularity will this be done? What would the query set (the set of unseen samples on which the influence of the data source is calculated) be?
- The position of paying by influence may end up being biased towards certain providers, which would defeat the purpose of the framework.

**Support:**

2

---

> ### Author Rebuttal · Authors · 2026-03-31
>
> We thank the reviewer for the insightful discussion and questions. We address the raised weaknesses and questions below.
>
> **W1 and Q2:**
> We define contribution primarily at the **data-sample level**, where each training sample is associated with an influence score on inference outcomes. This provides a natural and fine-grained unit for attribution that aligns with existing influence estimation methods.
>
> At the same time, our framework supports flexible aggregation across different granularities. In practice, contributions can be aggregated at higher levels, such as per dataset or per data provider. When a data provider contributes multiple samples (which is the typical case), their total contribution is obtained by accumulating influence across all associated samples.
>
> **W2 and Q3:**
> We agree that bias is an important concern. In stead of using raw influence scores, we can implement normalized and regularized attribution mechanisms to mitigate bias. For example, influence can be aggregated over diverse and representative query sets, normalized across providers, or combined with constraints such as caps, smoothing, or fairness-aware weighting to prevent dominance by a small subset of data sources.
>
> Bias mitigation can leverage existing approaches, including reweighting or rescaling of influence scores, distribution-aware normalization across data providers, robust aggregation (e.g., trimmed means or top-k truncation), and fairness constraints inspired by ranking and resource allocation systems. In addition, policy-level controls (e.g., minimum guarantees or contribution floors) can further balance incentives.
>
> More broadly, bias mitigation is a well-established and active research area, and addressing it fully is beyond the scope of this position paper. Such mechanisms can be implemented as an additional layer prior to the revenue allocation stage, without altering the core revenue-sharing model. In other words, bias handling is orthogonal to our framework and can be incorporated through existing or future techniques.
>
> We will expand the discussion in the revision to explicitly acknowledge bias concerns, provide concrete examples of mitigation strategies, and clarify how they integrate with our framework.
>
> **Q1:**
> Existing influence estimation methods (e.g., TracIn) demonstrate the feasibility of measuring training influence, and ongoing work continues to improve their efficiency (e.g., reducing gradient computations or using approximations). While such approaches may introduce modest accuracy trade-offs, they enable practical deployment.
>
> Importantly, our framework is method-agnostic and can incorporate more efficient alternatives as they emerge. Influence estimation can be performed offline or asynchronously, decoupled from the production system, and therefore does not impact inference latency.
>
> As a position paper, our goal is to outline a general architecture and highlight the importance of contribution-aware compensation. We also aim to encourage further research on scalable and efficient attribution mechanisms, which we view as a key enabler for broader adoption.
>
> **Q4:**
> The influence estimation for a given GenAI system is typically performed in a centralized manner, as it requires access to model internals and training data. In contrast, the revenue-sharing and auditing components can be implemented in a decentralized manner. Our framework supports both centralized and decentralized deployments depending on system and policy requirements.
>
> We further envision scenarios where a data provider contributes to multiple GenAI systems, and revenue contributions from different systems can be aggregated and settled through a shared (potentially decentralized) accounting layer, improving transparency and cross-system consistency.
>
> In practice, the system can operate fully in a centralized setting or adopt hybrid architectures, where influence is computed centrally while revenue allocation and auditing are handled in distributed or decentralized infrastructures.
>
> Decentralization offers several benefits, including reduced reliance on a single controlling entity, improved transparency and auditability (as attribution and revenue distribution can be verified by multiple participants), and increased trust among stakeholders by minimizing the risk of manipulation or dominance by any single party.
>
> We will clarify this separation of concerns and deployment flexibility in the revised version.
>
> **Q2 and Q3:**
> We have addressed Q2 and Q3 with our responses to W1 and W2, respectively.

---

> > ### Author Rebuttal · Reviewer_DaD3 · 2026-04-01
> >
> > - There have been recent works in the field that argue that sample-wise influence metrics for language modeling may not be the true measure for autoregressive tasks. Instead, they propose token-wise influence functions (at the cost of computation). Additionally, as the authors have mentioned, there are a few works that demonstrate the accuracy-efficiency trade-off, and the proposed framework is metric-agnostic.  My comment, however, is that such a trade-off must be discussed. For example, should the users of the framework focus more on the accuracy (which would mean the revenue is shared in a "fair" way) or could they sacrifice some of the accuracy for computation (in this case, the revenue being spent on computation is reduced, but the "fairness" may be biased), and should they focus on token-wise metrics (if so how does this change the interpretability)? I understand this is a loaded question, and the goal of this positional paper is to simply propose the idea of revenue sharing; however, should the paper be accepted, I feel it would be necessary for the readers of the paper to understand this. A few sentences/references that highlight this trade-off, and a few sentences on the authors' perspective of how much of the trade-off can be afforded, are needed in the paper.
> >
> > - My comment on the centrality of the framework also has a similar motivation. If the influence metrics were to be calculated in a centralized way, the "single point of failure/bias" comes into play. Also, who would pay for the centralized entity to compensate for the revenue on computation? Further, is it fair to call the system decentralized if the influence is calculated in a centralized fashion?  I understand that these are questions that may not have objective answers, but some discussion to ensure the readers understand the framework would be beneficial.
> >
> > - Additionally, I seemed to have missed highlighting some of the typographical errors in the paper in my initial review. Please remember to fix these errors in the revision (e.g., it's TracIn, not TrackIn, line 197)

---

### Official Review · Reviewer_tg3C · 2026-03-11

**Significance:** 3
**Argument Clarity:** 4
**Rating:** 5
**Confidence:** 4

**Questions:**

I have the following questions

1. Does everything need to be in revenue sharing?
2. How do we think public domain data needs to be handled? Does that pie go to the government? Or is it double taxation, since the profits from public domain works are already taxed by the government
3. Are there other/alternates view to the decentralized audit setting?
4. Would decentralized audits reveal private information about private companies, how can we appease adopters to adopt such systems?
5. Are there research bottlenecks that when solved can enable faster adoption of proposed positions apart from efficiency discussed in 6.1? Are multistep multi-environmental data attributions worth discussing in the paper? What about identifying data owners? Cases when multiple sources have the same data (often the case with copy pasted version of Wikipedia and such) While these are certain technically detailed questions, I urge the authors to consider updating 6.1.

**Alternative Views Section:**

Yes

**Compliance With Llm Reviewing Policy A Conservative:**

Affirmed.

**Discussion Potential:**

4

**Final Justification:**

The authors clarified the question I had and seems like there is consensus among reviewers so I feel justified in keep my original evaluation of 5.

**Paper Summary:**

The paper presents the position that data providers for gen-AI systems must be compensated via revenue sharing. This is broken down into 4 concrete positions. 1. Compensation is proportional to influence on output 2. Compensation should scale with the revenue/income of the corresponding gen-AI system 3. Compensation should be auditable 4. Moving in this direction needs community and policy support. They provide 2 alternates positions (a) no revenue sharing, i.e. under fair use (b) single time licensing.

**Position:**

Yes

**Position In Title:**

Yes

**Related Work:**

2

**Strengths And Weaknesses:**

## Strengths
1. Paper makes a strong case for revenue sharing, details why revenue sharing makes sense over the two alternates, i.e. the growth compounding from gen-AI system is not captured by either of the two alternate positions.

2. The paper clearly identifies the need for not only technical research development but also policy and community requirement, thus provides a strong discussion potential

3. Position 3 (decentralized auditability) is not a core position, but nevertheless a valuable position to include. Currently large web 2.0 platforms do not have such auditability and would be good to have such in the future.

## Weakness
1. Bridging the gap between technical readiness and policy. I think this is a particular weakness, firstly policy makers need immediate options, medium term policy options and long term policy options. Policy is going to have an outsize impact on the position of the paper. Thus more discussion on how to develop adaptable policy is needed. The reason I bring this up is discussed further in the next point.

2. Given the fast paced development, current gen-AI system specifically LLMs are no longer only pretrained but also RL trained in various environments, current attribution research is unable to handle this multi step training very well, more research is needed. Thus an initially policy could be a fast embedding similarity based attribution to training data can be the start of the policy recommendations, while the research catches up. Additionally the approaches suggested by the authors such as Tracin or others are still quite expensive, thus enforcing their use could make the business case of a lot of the gen-AI companies less economically viable. However I do concede that this is a "foster AI innovation vs protect data owners" boundary.

3. While decentralized is a strong approach, given the involvement of policy makers, decentralized is not necessary. Additionally I bring this up because of the public perception of the current blockchain technology. Often non technical issue arise (such as current reputation of a technology) when informing new policy decisons.

**Support:**

2

---

> ### Author Rebuttal · Authors · 2026-03-30
>
> We thank the reviewer for the insightful discussion and questions. We address the raised weaknesses and questions below.
>
> **W1:** We agree that bridging technical feasibility and policy is crucial. We will extend the paper with a phased policy pathway: (1) short-term: lightweight proxies (e.g., embedding similarity, dataset-level attribution) evaluated on subsets and possibly centralized systems; (2) medium-term: hybrid approaches with approximate attribution and partial auditing (e.g., sampled/top-k influence with auditability); (3) long-term: fully contribution-aware and auditable frameworks as methods mature (e.g., fuller realization of Figure 1). This enables policy to evolve with technical progress. We will update our discussion accordingly in the paper.
>
> **W2:** We agree that modern GenAI systems involve multi-stage pipelines (pretraining, fine-tuning, RLHF), and current methods do not fully capture cross-stage effects. Our framework supports compositional attribution across stages. In the near term, practical approximations (embedding-based, sampled, amortized) can balance cost and fidelity, while more efficient methods (e.g., improved gradient-based approaches) are an important research direction.
>
> More broadly, we agree this perfectly aligns with the “foster AI innovation vs. protect data owners” spectrum. Our goal is to provide a framework that enables a **win–win setting**, where data providers are fairly compensated while GenAI systems gain access to higher-quality and more sustainable data sources.
>
> **W3:**  We agree that decentralization is not a strict requirement and can be adopted at different levels. Building on W1, we envision a spectrum of deployment models: (1) centralized or minimally decentralized systems, with distributed components but centralized governance; (2) semi-decentralized systems (such as jointly maintained distributed databases with audit capabilities); (3) fully decentralized systems, where coordination is peer-to-peer and self-sustained. This flexibility supports real-world constraints and adoption.
>
> **Q1:** The short answer is no. The framework is selective and primarily targets copyright-protected data, where fair compensation is most relevant. It can be applied to chosen datasets/providers based on policy. The revenue-sharing component is decoupled from the GenAI pipeline as an auxiliary accounting layer as an incremental adoption.
>
> **Q2:** We sincerely thank the reviewer for this insightful question. How value derived from public-domain data should be handled (e.g., taxation, public funds, or no compensation) is a **policy question that varies across jurisdictions** (e.g., U.S., EU, Canada) and remains an open area of discussion. Our framework is designed to be **agnostic to these choices** and can coexist with different policy decisions without requiring changes to the underlying system.
>
> From our perspective, publicly available data (e.g., government records, court decisions) is generally intended to remain freely accessible and is typically not subject to copyright. As such, we do not consider it a primary target for revenue sharing in our framework at this stage.
>
> **Q3:** While decentralized auditing is one approach we highlight, it is not the only viable option. Our goal is to achieve auditability, and this can be realized through multiple designs, including centralized auditing by trusted operators, third-party or regulatory auditors, and hybrid or permissioned systems. As discussed in W1 and W3, we envision a spectrum of deployment models, ranging from centralized systems with distributed components, to semi-decentralized infrastructures (e.g., consortium-based or permissioned systems), and eventually to fully decentralized designs if appropriate. This flexibility allows the framework to adapt to practical constraints, policy requirements, and adoption considerations.
>
> **Q4:** Auditability does not require exposing sensitive information. Privacy-preserving mechanisms (aggregation, access control, selective disclosure) can reveal only derived metrics (e.g., contribution scores) rather than raw data. Permissioned systems, secure logging, and cryptographic proofs further protect confidentiality.
>
> **Q5:** Beyond efficiency (Section 6.1), several key research bottlenecks remain, including multi-stage attribution across complex pipelines (e.g., pretraining, fine-tuning, RLHF), data ownership and provenance (e.g., identifying rightful contributors and handling duplicated or widely shared content such as replicated web data). Our framework assumes identifiable data providers (as in current licensing-based systems), but extending to broader participation introduces challenges such as ownership verification and fraud prevention. These issues are not unique to our setting and are widely studied in data provenance and Web-scale systems. We will update Section 6.1 to explicitly discuss these challenges, including multi-stage attribution and data originality.

---

> > ### Author Rebuttal · Reviewer_tg3C · 2026-04-01
> >
> > The authors clarified the questions that were asked and provided actions that they would take to update the paper. I believe my assessment is fair and will keep the same score.

---

### Official Review · Reviewer_e1s5 · 2026-03-11

**Significance:** 3
**Argument Clarity:** 4
**Rating:** 5
**Confidence:** 3

**Questions:**

- How would you address or incorporate synthetic data in the proposed scheme?

**Alternative Views Section:**

Yes

**Compliance With Llm Reviewing Policy A Conservative:**

Affirmed.

**Discussion Potential:**

3

**Final Justification:**

The rebuttal clarified my concerns, which I reflect in my positive rating.

**Paper Summary:**

The paper argues for and outlines the benefits of a fairer revenue-sharing scheme that better and more transparently compensates at inference time (rather than some bulk compensation) the authors of the data used during model training. They also point out the characteristics such a system should have, like influence tracking, tokenized revenue indicators, and auditable smart contracts, and revenue-proportional payouts.

**Position:**

Yes

**Position In Title:**

No

**Related Work:**

2

**Strengths And Weaknesses:**

Strengths
- The manuscript is well written and easy to follow
- Alternative views and call-to-actions are explained clearly and in a fair fashion.

Weaknesses
- [Position not in the title] The title of the manuscript mentions the topic rather than the explicit position of the authors. From the title alone, it is not possible to understand what are the consequences of fair revenue sharing or why it is needed. I argue a title along the lines of “Fair revenue sharing improves quality of available data” would have been more clear.
- [Position 2] The authors claim that “Revenue-proportional compensation aligns incentives across stakeholders.”. However, I do not see why this is the case. I presume this could be the case as the data-owner would be willing to share data only if it retains the revenue sharing fair. If this is the case, I argue this is not clearly explained in the manuscript.
- [Ease of attribution] The authors claim that such an attribution system would not be complex enough to slow down technical progress of ML systems. In support of this statement they bring the example of other ecosystems such as Spotify (line 198, right column). However, ecosystems such as Spotify are severely different from the ones of GenAI, as the revenue attribution is straight-forward: a user listening to a song gives revenue to who owns licensing rights of such song. Datasets of modern GenAI systems contain data samples coming from millions of users. Attributing the outcome of a query to a sample is not as straight-forward.
- [Synthetic data to circumvent revenue attribution] Nowadays synthetic data is more and more common, even for training ML models. With the proposed protocol, I wonder whether one party could just train a synthesizer with the data given by the data provider. Then, use this synthesizer to generate data to train models that are actually deployed. Hence, circumventing the proposed revenue scheme.

**Support:**

3

---

> ### Author Rebuttal · Authors · 2026-03-30
>
> We thank the reviewer for the constructive feedback. We address the concerns below.
>
> **W1:** We thank the reviewer's suggestion and will revise the title to: “Position: From Fair Use to Fair Share: GenAI Systems Should Implement Fair Revenue Sharing for Data Providers.”
>
> **W2:** Our argument is that revenue-proportional compensation aligns incentives by linking rewards to the actual economic value generated at inference time. In the current regime, data providers are either unpaid or compensated through fixed, one-time agreements that are disconnected from downstream value. In contrast, revenue-proportional schemes incentivize data providers to contribute higher-quality data, model developers to improve attribution accuracy and transparency, and the ecosystem to sustain long-term data availability. More broadly, fair compensation encourages data providers to share and curate high-quality data tailored for model training, strengthening the overall ecosystem. We will clarify this incentive alignment more explicitly in the revision.
>
>
> **W3:** We thank the reviewer for this important clarification. Our intent is not to claim that attribution in GenAI is as simple as in systems like Spotify. Rather, we argue that the **economic model of compensating data providers should not hinder system development**. In fact, fair compensation can incentivize data providers to supply higher-quality data, improving the overall ecosystem. We do not require exact attribution; instead, our framework supports approximate and scalable methods (e.g., TrackIn, UnTrack, and DataInf), which are active research areas. Moreover, the attribution, revenue calculation, and compensation components are **decoupled from the GenAI model** and can be **computed offline or asynchronously**, thus not impacting inference latency. We will clarify that the Spotify example refers to revenue-sharing mechanisms, not attribution complexity.
>
> **W4 and Q1:** We agree that synthetic data introduces challenges for attribution. However, synthetic data does not eliminate contribution: it transforms it. If a synthesizer is trained on original data, its outputs remain statistically dependent on that source. In our framework, attribution can be extended to capture upstream influence across multi-stage data pipelines, assigning partial credit to original data providers.
>
> That said, if an original provider’s data significantly influences Model A, and Model A generates synthetic data that in turn influences Model B, then the original provider should still receive partial credit. Synthetic data should not erase attribution; rather, **it should inherit and redistribute contribution**. In this setting, influence can be computed via compositional attribution across stages, where the contribution of original data to Model B’s outputs is approximated by propagating influence through the synthetic data generation process.
>
> For example, if provider P contributes 20% of Model A’s effective training influence, and synthetic data from Model A accounts for 30% of Model B’s behavior on a workload, then P receives a proportional share of that 30% contribution (e.g., ~6%), possibly adjusted by normalization or caps. In practice, this can be implemented at the dataset or batch level (rather than per-sample) using lineage tracking, approximate influence composition, and generation logs.
>
> We note that synthetic data generation is often performed within GenAI pipelines by model operators (e.g., in distillation [1, 2]), in which case it is already part of the system and does not introduce additional compensation requirements. For external generators, contributions can still be tracked through data lineage and provenance mechanisms. In addition, governance at the data ingestion and provenance layer (e.g., requiring source disclosure or restricting untracked data) can make circumvention detectable and auditable.
>
> We view robust attribution under synthetic data and multi-stage pipelines as an important research direction rather than a fundamental limitation, and will clarify this extension in the revision.
>
> [1] On Efficient Distillation from LLMs to SLMs, NIPS’24
> [2] Domain-Specific LLM Adaptation: Bridging Personalization and Efficiency Through Synthetic Data and Optimization, AAAI’26

---

> > ### Author Rebuttal · Reviewer_e1s5 · 2026-04-04
> >
> > Thanks for the rebuttal. I have no further questions.

---

### Official Review · Reviewer_spfa · 2026-03-11

**Significance:** 4
**Argument Clarity:** 3
**Rating:** 5
**Confidence:** 3

**Questions:**

1) You argue that revenue generated on the side of the GenAI model operators should be shared. What about revenue that is generated beyond that? One could think of end users who use model outcomes to generate value for their own businesses. This might be partially covered by a subscription fee paid by this end-user, but this does not really scale with the amount of revenue generated.
2) Is advertising revenue something that is currently applied anywhere?
3) What are the monetary funds that you mention in line 63? Would these constitute a potential third alternative view?

**Alternative Views Section:**

Yes

**Compliance With Llm Reviewing Policy A Conservative:**

Affirmed.

**Discussion Potential:**

4

**Final Justification:**

My initial assessment of the paper was borderline, with a positive tendency. As all my concerns and questions were satisfactorily addressed and the authors promised to improve the paper based on the reviews, I am leaning even more toward the positive. Hence, I am giving a clear valuation of 5.

**Paper Summary:**

The paper addresses how data providers should be compensated for their contributions to revenue generated by AI systems.
The authors' main position is that generative AI systems should apply revenue sharing mechanisms that compensate data providers proportionally to their contribution to the model outputs at inference time.
Quality of data used in training has an impact on the quality of the outputs of an AI system, which in turn directly translates to revenue collected through subscription fees, advertising income, and the like. From this observation, the authors derive the need for fair revenue sharing.

The authors start by presenting the two payment mechanisms that are actively used: no compensation for data providers at all (e.g., under the fair use doctrine in the US), or compensation agreed upon upfront through deals between data providers and GenAI model providers (e.g., licensing or partnership agreements). They argue that the first option poses an existential threat to the creators, discourages the creation of high-quality content, and leads to legal tensions, while the second is mostly a choice for larger data providers, leaving out smaller ones. There does not exist a fair, large-scale, verifiable, and proportional mechanism that is widely adopted.

According to the authors, this state of affairs might lead to the following risks: (1) freely available data might become less and less available, due to increasing numbers of content creators/holders that restrict access to their data, while at the same time, training on high-quality data is important for model quality. (2) The highest-quality data is most sought-after, and simultaneously often the most protected, which leads to a bottleneck situation. (3) Ongoing legal uncertainty might lead to investment reluctance. (4) Jurisdiction pertaining to this topic is fragmented across the globe. This might incentivize GenAI developers to deploy debatable workarounds, which in turn undermines trust.

These premises lead to the aforementioned main position. For their argumentation, the authors split their position into multiple sub-positions.
Position 1: Data providers should receive compensation proportional to the impact they have on the outcome of a GenAI system at inference time.
Position 2: Compensation should scale with the amount of generated revenue.
Position 3: The compensation framework should be auditable and verifiable by all participants.
Position 4: Successful real-world adoption needs acknowledgement and support from the community, industry, and legal and regulatory authorities.

The authors propose a framework that aims to achieve all of these claims: Influence is tracked and translated to a token currency. The value of these tokens can be scaled with the revenue, independently of how they are attributed. The compensation is managed through smart contracts on blockchains that can be audited by all authorized participants.

Next, the two alternative views that either uncompensated fair use or one-time licensing is sufficient are discussed.
Finally, the authors enumerate what still needs to be done to achieve the presented goals in the Call to Action section.

**Position:**

Yes

**Position In Title:**

No

**Related Work:**

3

**Strengths And Weaknesses:**

Strengths:
- This is a mostly clearly written paper that follows a thorough argumentation: First, they state the current state and identify problems and potential risks caused by it. Next, they clearly explain their positions and subsequently propose an appropriate framework. They complete their argumentation by discussing alternative views and identifying action items for researchers, authorities, and the community to eventually achieve the goals.
- The abstract identifies the paper as a position paper and clearly states the position.
- The introduction states the position in bold text.

Weaknesses:
- The title does not comply with the instructions. It does not clearly —rather, implicitly —formulate a position. I would instead expect something like "GenAI Systems Should Implement Fair Revenue Sharing for Data Providers" or similar.
- Some concepts are not explained. Examples include: the concept of advertisement revenue (What exactly  is it? Where does the revenue come from? Is it based on a contract? Is it currently used anywhere?); smart contracts; how is the blockchain integrated into the framework? How is quality defined/measured?
- Evidence is missing to support the claim that quality of the data and quality of the outputs are correlated. Merely, in lines 158-161, it is stated that "High-quality data sources [...] are widely viewed as particularly valuable for model capability and reliability". Please provide more specific evidence or mark this as speculation.
- Some argument is missing that links content quality and revenue. For advertisement-based revenues, I can see this link if you equate input quality with output quality, and output quality with the amount of revenue generated through the output directly. But in general, this does not seem to be that straightforward. Consider, for example, subscription fees. These provide constant revenues that do not directly depend on the model's outputs. That is, in a system where all revenue is generated through subscription fees only, output quality cannot be equated with revenue. Instead, individual revenue would rather scale with the frequency and extent to which the content influences model outputs. This, however, does not necessarily seem connected to quality, in my opinion. For example, think of a dialog in which a user repeatedly asks an LLM for something because they do not receive the answer they are hoping for. Eventually, the LLM delivers a satisfying output. How would revenue attribution work in this case? Would revenue be split proportionally between the sources that impacted the (intermediate) answers? Would the satisfying answer (that arguably leads the user to keep paying the subscription fee) generate higher revenues (if this answer can even be identified)?
-> That said, I think there is some room for debate about whether this is an appropriate or feasible solution for the aforementioned problems. I think the limitations and potential pitfalls of the proposed model could be handled more transparently in the paper.
- On page 6, the proposed revenue-sharing framework is demonstrated using an example. However, it only considers the case of advertising revenue, which, if I am not mistaken, might still be a theoretical concept not applied in practice at this point.

_Minor remarks_
- The exact same word-for-word quotation appears twice in lines 58-62 and then again in lines 120-123 (right half of the page)
- Al attribution techniques mentioned in lines 194-196 should be equipped with a citation (or at least the references given at the end of the sentence should be matched to the corresponding techniques).
- The percentages in the example on page 6 (accidentally?) sum to 100%. If I understand correctly, the two users and consequently the generated revenues should be independent of each other. It might be a good idea to change the values slightly to prevent the potential misunderstanding that the numbers should add to 100.
- In the conclusion: I find the part of the sentence "[...] and showed that this is a systems problem, not merely a legal or ethical one" ambiguous. What do you mean by 'systems problem'? And where do you show this?

**Support:**

2

---

> ### Author Rebuttal · Authors · 2026-03-30
>
> We thank the reviewer for the detailed and constructive feedback. We address the concerns and questions below.
>
> W1: We thank the reviewer's suggestion and will revise it to: “Position: From Fair Use to Fair Share: GenAI Systems Should Implement Fair Revenue Sharing for Data Providers.”
>
>
> W2: Advertisement revenue in LLM-based systems is analogous to ads in internet platforms (e.g., Google and Meta), where revenue is generated through user interactions such as clicks, impressions, or conversions. (Please also see our response to Q2).
>
>
> W3: We will provide stronger evidence to support this claim, which is widely recognized in the ML literature. The cited works [Duan’25; Villalobos’24] (at the end of the referred sentence) already show that LLM performance depends critically on training data quality. We will further strengthen this by adding more explicit evidence: [1] states that ML system performance strongly depends on data quality, and [2] shows that poor-quality data leads to inaccurate and unreliable models. We will incorporate these citations and clarify the claim.
>
> [1] A Survey of Data Quality Requirements That Matter in ML Development Pipelines
> [2] Data Readiness for AI: A 360-Degree Survey
>
>
> W4: We agree that the relationship between output quality and revenue may not be always direct. However, **our framework is designed to adapt to different revenue calculation methods through a modular (“plug-in”) revenue indicator**. For example, in subscription-based settings, GenAI systems can estimate contribution using user satisfaction signals, retention metrics, or platform-specific evaluation algorithms. Rather than assuming per-output attribution, we can aggregate contribution over sessions or longer horizons, capturing user-perceived value (e.g., satisfaction, retention). In multi-turn interactions, attribution can be aggregated at the session level rather than assigning equal weight to intermediate responses, thereby aligning rewards with final utility. The revenue indicator can be instantiated differently depending on the monetization model.
>
> For advertisement-based models, the calculation is more direct: the clicks and conversions of proposed link dominates the calculation. We will extend the discussion of subscription-based revenue in both the position and the example in the final version, and clarify these assumptions and limitations.
>
>
> W5: Extending our response for W3, we will extend the current ad-based example to also cover subscription-based scenarios and clarify that the framework supports both monetization models. Please also see our response to Q2 for additional details.
>
> Due to space limitations, we do not address minor remarks individually, but we will revise all noted issues (duplicate text, missing citations, example clarity, wording).
>
>
> Q1: We thank the reviewer for this important point. We agree that significant value can be generated downstream by end users. In this work, **we intentionally scope compensation to revenue captured by GenAI operators** (e.g., subscriptions, usage fees, advertising), as this is the **most measurable and enforceable layer** without requiring visibility into external business operations. Extending compensation to downstream value chains would require tracking how model outputs propagate into external systems, raising challenges in observability, privacy, and enforcement. We view this as an important future direction (e.g., usage-based royalties or API-level reporting), with platform-level revenue sharing as a necessary first step.
>
>
> Q2: Yes, advertising-based monetization is already emerging in GenAI systems in multiple forms, including sponsored responses, affiliate links, and conversational ads integrated into outputs. For major platforms, Google has already begun integrating advertising into AI-powered search experiences, and is actively testing advertising capabilities within Gemini (https://www.wired.com/story/google-nick-fox-advertising-search-ai-gemini). More broadly, **this reflects a clear industry trend: advertising remains the dominant revenue model for large internet companies** (e.g., ~77% of Google’s and ~98% of Meta’s revenue comes from ads). Therefore, in our humble view, incorporating advertising into GenAI systems is no longer speculative but a natural extension of existing monetization strategies.
>
>
> Q3: We clarify that “monetary funds” refer to compensation schemes such as collective funds or pooled payments for data providers. This approach can be viewed as a variant of Alternative View 2 (licensing-based approaches), as it relies on predefined payments rather than contribution-aware, inference-time attribution. We will clarify this categorization in the revision.
>
> We will (1) revise the title to explicitly state the position, (2) add more citations supporting data quality claims, (3) clarify the relationship between contribution and revenue and expand examples to include subscription models, and (4) improve raised minor issues.

---

> > ### Author Rebuttal · Reviewer_spfa · 2026-04-02
> >
> > Thank you for your thorough reply. All my questions are resolved. Personally, I think your answers to questions 1 and 2 are interesting and noteworthy to provide a better contextualization. After revising the paper as promised here and in the other review responses, I think it will provide a sound overview and a solid foundation for further discussion. Therefore, I will increase my score to 5.

---

### Decision · Program_Chairs · 2026-04-30

**Decision:**

Accept (regular)

**Comment:**

The reviewers acknowledge that the paper presents a well-structured, and clearly articulated position advocating for contribution-aware, revenue-proportional compensation in GenAI systems. Its strengths lie in the logical progression of arguments, balanced discussion of alternative views, and inclusion of technical, policy, and ecosystem-level considerations. The framework is seen as conceptually strong and relevant, with meaningful implications for future research and governance.

While several weaknesses were identified, such as limited explanation of key concepts, insufficient empirical support for certain claims, challenges in attribution feasibility, and gaps between technical proposals and policy readiness. The authors’ rebuttal has satisfactorily addressed most concerns. In particular, clarifications around incentive alignment, acknowledgment of technical limitations, and expanded discussion of practical constraints improved the paper significantly.

Overall, despite remaining open challenges, reviewers agree that the paper makes a valuable and timely contribution to  fair data compensation in GenAI. Following rebuttal, the consensus is to accept the paper.